# One Model to Edit Them All: Free-Form Text-Driven Image Manipulation with Semantic Modulations

**Yiming Zhu**[1*]    **Hongyu Liu**[2*]    **Yibing Song**[3†]    **Ziyang Yuan**[1]    **Xintong Han**[4]

**Chun Yuan**[1†]    **Qifeng Chen**[2]    **Jue Wang**[3]

[1]Tsinghua Shenzhen International Graduate School
[2]Hong Kong University of Science and Technology
[3]Tencent AI Lab    [4]Huya Inc
zym20@mails.tsinghua.edu.cn    hliudq@cse.ust.hk
yibingsong.cv@gmail.com    yuanc@sz.tsinghua.edu.cn

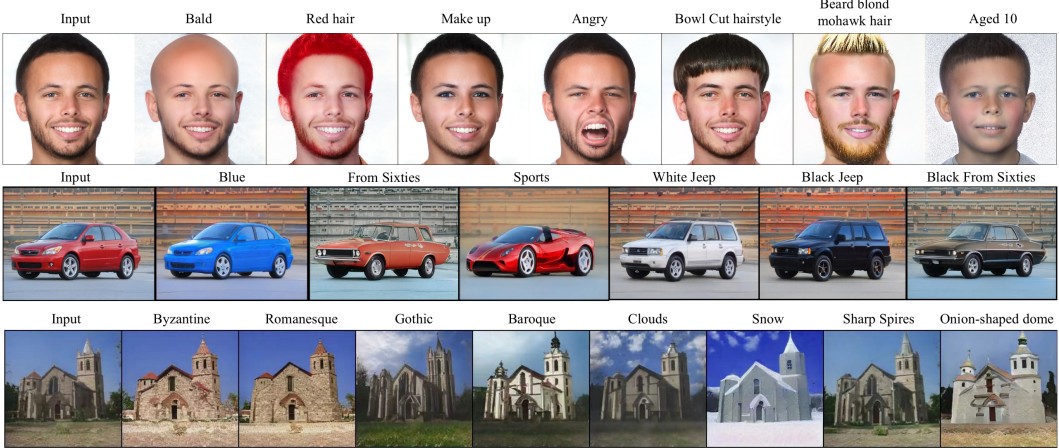

Figure 1: The proposed FFCLIP model edits each type of image with free-form text prompts. For each input image shown on the first column, we show manipulation results with text prompts on their corresponding row. Each text prompt, which contains a single semantic meaning (e.g., 'Blue') during training, can convey multiple semantics (e.g., 'Black from sixties' and 'Beard bond mohawk hair') during inference for free-form image manipulations.

## Abstract

Free-form text prompts allow users to describe their intentions during image manipulation conveniently. Based on the visual latent space of StyleGAN [21] and text embedding space of CLIP [34], studies focus on how to map these two latent spaces for text-driven attribute manipulations. Currently, the latent mapping between these two spaces is empirically designed and confines that each manipulation model can only handle one fixed text prompt. In this paper, we propose a method named Free-Form CLIP (FFCLIP), aiming to establish an automatic latent mapping so that one manipulation model handles free-form text prompts. Our FFCLIP has a cross-modality semantic modulation module containing semantic alignment and injection. The semantic alignment performs the automatic latent mapping via linear transformations with a cross attention mechanism. After align-

---

*Y. Zhu and H. Liu contributes equally. †Y. Song and C. Yuan are corresponding authors. This work is done when Y. Zhu is an intern in Tencent AI Lab.

ment, we inject semantics from text prompt embeddings to the StyleGAN latent space. For one type of image (e.g., 'human portrait'), one FFCLIP model can be learned to handle free-form text prompts. Meanwhile, we observe that although each training text prompt only contains a single semantic meaning, FFCLIP can leverage text prompts with multiple semantic meanings for image manipulation. In the experiments, we evaluate FFCLIP on three types of images (i.e., 'human portraits', 'cars', and 'churches'). Both visual and numerical results show that FFCLIP effectively produces semantically accurate and visually realistic images. Project page: https://github.com/KumapowerLIU/FFCLIP.

# 1   Introduction

Neural image synthesis has received tremendous investigations since the Generative Adversarial Networks (GANs) [14]. The synthesized image quality is significantly improved via the StyleGAN-based approaches [22, 21, 20]. Recently, free-form text prompts describing user intentions have been utilized to edit StyleGAN latent space for image attribute manipulations [33, 53]. With a single word (e.g., 'Blue') or phrase (e.g., 'Man aged 10') as an input, these methods edit the described image attribute accordingly by modulating the latent code in StyleGAN latent space.

The accurate image attribute manipulation relies on the precise latent mapping between the visual latent space of StyleGAN and the text embedding space of CLIP. An example is when the text prompt is 'Surprise,' we first identify its related semantic representations (i.e., 'Expression') in the latent visual subspace. Then, we modulate the latent code of this identified latent subspace via the text embedding guidance. Pioneering studies like TediGAN [53] and StyleCLIP [33] empirically identify which latent visual subspace corresponds to the target text prompt embedding (i.e., attribute-specific selection in TediGAN and group assignment in StyleCLIP). This empirical identification confines that given one text prompt, they must train a corresponding manipulation model. Different text prompts require different manipulation models to modulate the latent code in latent visual subspace of StyleGAN. Although the global direction method in StyleCLIP does not employ such a process, the parameter adjustment and edit direction are manually predefined. To this end, we are motivated to explore how to map text prompt embeddings to latent visual subspace automatically. So a single manipulation model is able to tackle different semantic text prompts.

In this paper, we propose a free-form method (FFCLIP) that manipulates one image according to different semantic text prompts. FFCLIP consists of several semantic modulation blocks that take the latent code $w$ in StyleGAN latent space $\mathcal{W}^+$ [1] and the text embedding as inputs. Each block has one semantic alignment module and one semantic injection module. The semantic alignment module regards the text embedding as the query, the latent code $w$ as the key, and the value. Then we compute the cross attention separately in both position and channel dimensions to formulate two attention maps. We use linear transformations to perform a latent mapping from text prompt embedding and latent visual subspace, where the linear transformation parameters (i.e., translation and scaling parameters) are computed based on these two attention maps. Through this alignment, we identify each text prompt embedding to its corresponding StyleGAN latent subspace. Finally, the semantic injection module modifies the latent code in subspace via another linear transformation following HairCLIP [51]. The modulated semantics are represented as the latent code offset $\Delta w$, which is refined progressively through several semantic modulation blocks.

From the perspective of FFCLIP, the empirical group selection of $w$ in [33, 51] is a particular form of our linear transformations in the semantic alignment module. Their group selection operations resemble a binary value of our scale parameters to indicate the usage of each position dimension of $w$. On the other hand, we observe that the $\mathcal{W}^+$ is not disentangled completely, the empirical design could not find the precise mapping between StyleGAN's latent space and CLIP's text semantic space. In contrast, the scale parameters in our semantic alignment module adaptively modify the latent code $w$ to map different text prompt embeddings. The alignment is further improved via our translation parameters. We evaluate our method on the benchmark datasets and compare FFCLIP to state-of-the-art methods. The results indicate that FFCLIP is superior in generating visually pleasant content while conveying user intentions.

## 2 Related Works

**Latent Space Image Manipulation.** There are a wide range of image manipulation and restoration studies in the literature [29, 31, 30] and we focus on discuss the StyleGAN based methods here. The latent space in StyleGAN [22, 21, 20, 26] has demonstrated great potential in representing the semantics in the image, motivating many works to disentangle the latent space for controllable image manipulation. Specifically, investigations [40, 18, 3, 45, 4] predict the meaningful offsets or directions in the latent space given image annotations as supervision, while studies [41, 48, 16, 49] disentangle the latent space in an unsupervised manner to find the semantic directions. Although these methods achieve great performance in latent space manipulation, they can only find limited semantic directions that confine user intentions. In contrast, we achieve free-form image manipulation conditioned on arbitrary text prompts, giving users more degrees of editing freedom. Our StyleGAN inversion encoder is related to GAN inversion methods [1, 2, 5, 6, 38, 46, 50] that maps the image to the latent code space $w \in \mathcal{W}^+$ [1]. In our work, we use the e4e [46] as StyleGAN inversion encoder following StyleCLIP [33], and introduce the cross attention mechanism [47, 13, 28] to linearly map the text and visual latent space for free-form editing.

**Text-driven Image Generations.** Starting from [37] that leverages text embeddings as the condition for GAN-based image training, several works [56, 57, 58] improve the synthesis quality by introducing multi-scale or hierarchically-nested GANs. And attention modules are introduced in [54] to match the generated images to text. [8, 32, 27, 44] improves image content fidelity is considered from the network structure and training perspectives. The text to image generation performance is further boosted in DALLE [36, 35] and DiffusionCLIP [23] where text prompts with multiple semantic guidance are translated into images. Vision transformer [10] and diffusion model [15] also show impressive results in text-driven image manipulations. TediGAN [53] transfers the image and text to a shared StyleGAN latent space, and modulates the image latent vector with text. Recently, CLIP [34] constructs a text-image embedding space to connect the semantics between image and text embeddings. Combining the CLIP model with StyleGAN becomes promising for text-driven image manipulations. Pioneering works such as StyleCLIP [33] use the CLIP model to map semantics from text to images. HairCLIP [51] builds on top of StyleCLIP while focusing more on the hair regions with reference images and semantic injections. FEAT [17] proposes an attention mask to prevent changes to unedited areas . These methods need to manually select a specific StyleGAN latent subspace according to the target text. So they fail to train a single model for different text prompts. In comparison, FFCLIP can adaptively match one text embedding to the desired latent subspace, making one model process multiple text prompts for the corresponding image manipulations.

## 3 Proposed Method

Fig. 2 shows an overview of the proposed method. Given an input image and a text prompt, we can obtain the StyleGAN latent code $w$ and text embedding $e_t$. We propose $k$ (we set $k = 4$ in practice) semantic modulation blocks where each block first aligns semantics between $w$ and $e_t$ and then edits the latent code. The output of each block is an offset $\Delta w$ which adds to the latent code $w$. Finally, a StyleGAN generator decodes the resulting latent code to the manipulated image. In the following, we first briefly review the StyleGAN and the CLIP model, then we illustrate how we align and inject semantics for latent space editing.

**StyleGAN Revisiting.** Studies for StyleGAN [22, 21, 20] generate visually realistic face images with various semantics. In StyleGAN, a latent variable $\mathcal{Z} \in \mathbb{R}^{512}$ from the Gaussian distribution is gradually transformed to a disentangled latent code $\mathcal{W} \in \mathbb{R}^{512}$ with semantic meanings. The latent code $\mathcal{W}$ is then fed into 18 layers for image generation. Later, investigations [46] find that an extension of $\mathcal{W}$ (i.e., $\mathcal{W}^+ \in \mathbb{R}^{18 \times 512}$) is more effective for identity preservation when manipulating the image . There is a definite relation between the layers of $\mathcal{W}^+$ and the semantic latent subspace. So the semantic modulation can be formulated as editing some specific layers of $\mathcal{W}^+$. As a result, image manipulation methods [3, 45, 4, 18] based on StyleGAN invert an input image to a latent code $w \in \mathcal{W}^+$ for semantic modification. We follow [33] to utilize the StyleGAN2 as our generator.

**CLIP Revisiting.** The CLIP model [34] is pretrained from 0.4 billion image-text pairs for cross-modality semantic matching. This model consists of an image encoder and a text encoder to project an image and a text prompt into a 512-dimension embedding, respectively. We leverage the text

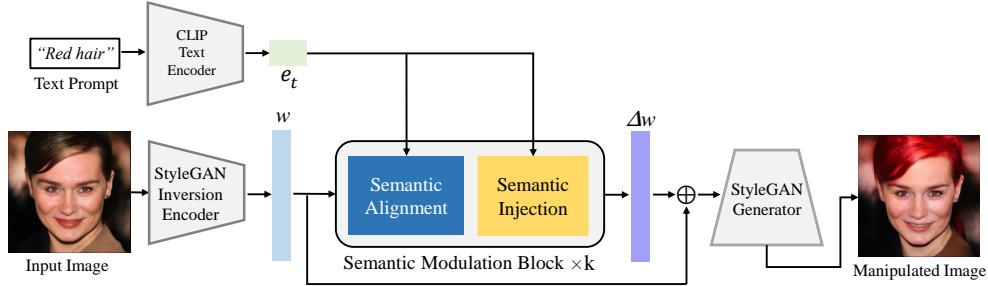

Figure 2: Overview of our pipeline. Given a text embedding $e_t$ and an input latent code $w$, we propose several semantic modulation blocks to produce an offset $\Delta w$ for latent space modification. In each block, there is a semantic alignment module to automatically map the semantic subspace in $w$ based on $e_t$. Then, the semantic injection module modifies this latent subspace to refine $\Delta w$.

encoder of CLIP model to produce the text prompt embedding. This encoder is versatile in perceiving text prompts with different semantic meanings.

## 3.1 Semantic Modulation

Fig. 3 shows the framework of a semantic modulation block where there are $k$ blocks in total. The inputs are an image latent code $w \in \mathcal{W}^+$ converted from an image with e4e [46], and a text embedding $e_t$ extracted from the CLIP model. We use $k$ semantic modulation blocks progressively refine $\Delta w$. The $\Delta w$ will modify $w$ for latent subspace editing. The output of the $i$-th block is denoted as $\Delta w_i$. In each block, there is a semantic alignment module followed by a semantic injection module, which are illustrated as follows:

**Semantic Alignment Module.** The automatic semantic correspondence between $e_t$ and $w$ enables that one model is able to process text prompts with different semantic meanings. We establish this automatic correspondence by aligning $w$ and $e_t$ via a linear transformation. The parameters of this linear transformation are learned based on the cross attention measurement. Specifically, for the semantic alignment module in the $i$-th block, we compute a cross attention map between $e_t$ and $\Delta w_{i-1}$ ($\Delta w_0 = w$) in both position and channel dimensions similar to the dual attention network [11]. The scale and translation parameters of the linear transformation are computed based on the cross attention maps.

For the position dimension, we set the $e_t$ as the Query $Q_p \in \mathbb{R}^{512}$ and the latent code $\Delta w_{i-1}$ as the Value $V \in \mathbb{R}^{18 \times 512}$ and the Key $K \in \mathbb{R}^{18 \times 512}$. Then, we compute the cross attention map Attention$_p$ as a scale parameter $S \in \mathbb{R}^{18}$. The scale parameter $S$ models the Value $V$ in position dimension for transforming the latent space to match the text embedding semantics. The cross attention computation on the position dimension can be written as:

$$\begin{aligned} Q_p &= e_t W^Q, K = \Delta w_{i-1} W^K, V = \Delta w_{i-1} W^V, \\ \text{Attention}_p &= \text{Softmax}(Q_p K^T), \\ S &= \text{Attention}_p, \end{aligned} \tag{1}$$

where $W^Q, W^K, W^V \in R^{512 \times 512}$. We use these scale parameters to adjust the contribution of each dimension of $\mathcal{W}+$ conditioned on $e_t$. Meanwhile, we observe that the $\mathcal{W}^+$ space is not fully disentangled. One semantic property is reflected not only in a specific element of $w$, but in other elements as well. To this end, we compute the translation parameters in the channel dimension to enhance the semantic alignment.

For the channel dimension, we use the same $K$ and $V$ as those in the position dimension, and a new Query $Q_c \in \mathbb{R}^{18 \times 512}$ to calculate the cross attention map Attention$_c$ in the channel dimension. Then, we reconstruct the Value V with the attention map Attention$_c$ and utilize an adaptive average pooling

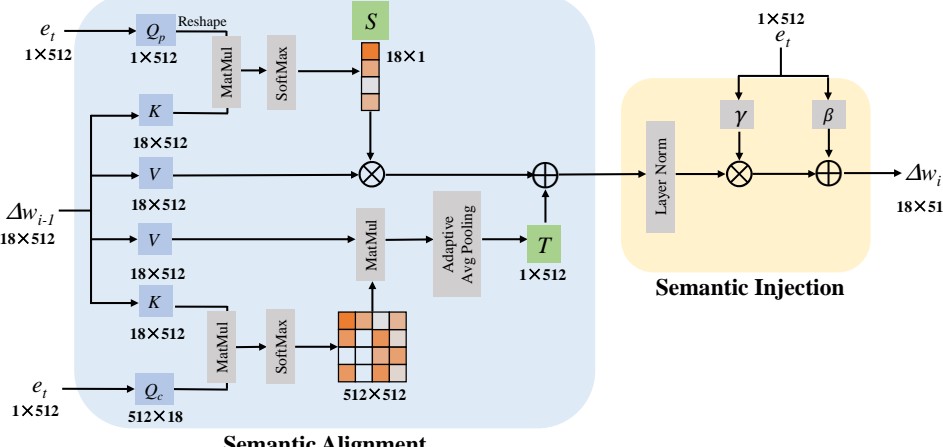

**Semantic Alignment**

**Semantic Injection**

Figure 3: Our semantic modulation block consists of a semantic alignment module and a semantic injection module. Given $\Delta w_{i-1}$ and $e_t$ as input, we compute their cross attentions in position and channel dimensions to learn a linear transformation. This transformation aligns $\Delta w_{i-1}$ to $e_t$. Then, we use semantic injection to refine $\Delta w_{i-1}$ to $\Delta w_i$ for the input to the next block.

to compute the translation parameters $T \in \mathbb{R}^{512}$. These processes can be written as

$$
\begin{aligned}
Q_c &= e_t^T W_c^Q, K = \Delta w_{i-1} W^K, V = \Delta w_{i-1} W^V, \\
\text{Attention}_c &= \text{Softmax}(Q_c K), \\
T &= \text{AAP}(\text{Attention}_c V),
\end{aligned}
\tag{2}
$$

where the $W^K$ and $W^V$ are the same as the those in the position dimension, and the $W_c^Q \in R^{1 \times 18}$. AAP is the adaptive average pooling operation. After we get the scale and translation parameters, we can align $\Delta w_{i-1}$ to $e_t$ with a linear transformation as follows:

$$
x_i = S \times V + T,
\tag{3}
$$

where the $x_i$ is the output of this semantic alignment module. Then, we inject $e_t$ to this aligned latent subspace via our semantic injection module in the following.

**Semantic Injection Module.** In this module, we inject the semantic from $e_t$ to $x_i$ following that in HairCLIP [51]. Specifically, we adopt the fully-connected layers to map the text embedding $e_t$ into two injection parameters $\beta \in \mathbb{R}^{512}$ and $\gamma \in \mathbb{R}^{512}$, respectively. Then, we inject the text embedding semantics as follows:

$$
\Delta w_i = (1 + \gamma) \frac{x_i - \mu_{x_i}}{\sigma_{x_i}} + \beta,
\tag{4}
$$

where $\mu_{x_i}$ and $\sigma_{x_i}$ represent the mean and the variance of the input $x_i$, respectively. With the aligned semantics, our injection produces the $\Delta w_i$ to modify StyleGAN latent code. The $\Delta w_i$ is further refined through the following semantic modulation blocks to obtain $\Delta w$ for the final image generation.

## 3.2 Training Objectives

In text-driven image manipulation, we focus on two aspects of the output result. First, the semantics of the target object in the image shall be consistent after manipulations. Second, the output image shall be semantically relevant to the text prompt. To this end, we follow [51] and develop two types of loss functions, i.e., semantic preserving loss and text manipulation loss.

**Semantic Preserving Loss.** We aim to preserve the semantic consistency between the input and output images. This semantic shall be represented within the CLIP model. Specifically, we feed the input and output images through the CLIP image encoder to get two image embeddings. If the image

subject is consistent, the distance between these two embeddings should be small. So we reduce the distance between these two embeddings. We call it as image embedding loss and define it as follows:

$$\mathcal{L}_{embd} = 1 - cos\{E_I^{CLIP}(G(w')), E_I^{CLIP}(G(w))\} \tag{5}$$

where the $G(\cdot)$ denotes the pretrained StyleGAN generator, $cos\{\cdot\}$ means cosine similarity, $E_I^{CLIP}(\cdot)$ is the pretrained CLIP image encoder, $w$ and $w' = w + \Delta w$ are the input and our edited StyleGAN latent codes, respectively. Then we utilize the $L_1$ norm for preserving the irrelevant semantic regions:

$$\mathcal{L}_{norm} = \|\Delta w\|_1. \tag{6}$$

Moreover, for 'human portrait' images, the background loss and the face identity loss are used to improve the performance, and the face identity loss is defined as follows:

$$\mathcal{L}_{id} = 1 - cos\{R(G(w')), R(G(w))\}, \tag{7}$$

where $R(\cdot)$ indicates the pretrained ArcFace network [9]. Also, the background loss can be formulated as:

$$\mathcal{L}_{bg} = \|(G(w') - G(w)) * (P(G(w')) \cap P(G(w)))\|_2 \tag{8}$$

where $P(\cdot)$ is the facial parsing network [25] and $P(G(w'), P(G(w))$ represents the non-facial regions in the input and output images, respectively. The overall semantic preserving loss $\mathcal{L}_{sp}$ can be written as

$$\mathcal{L}_{sp} = \lambda_{embd} \cdot L_{embd} + \lambda_{norm} \cdot L_{norm} + \lambda_{id} \cdot L_{id} + \lambda_{bg} \cdot L_{bg}, \tag{9}$$

where $\lambda_{embd}, \lambda_{norm}, \lambda_{id}$, and $\lambda_{bg}$ are the weights that adjust the contribution of each loss term. We set $\lambda_{embd} = 1.0, \lambda_{norm} = 1.5, \lambda_{id} = 1.0$, and $\lambda_{bg} = 2.0$. In particular, we only introduce $\mathcal{L}_{id}$ and $\mathcal{L}_{bg}$ when we edit face images.

**Text Manipulation Loss.** To evaluate the correlation between output images and text prompt embeddings, we minimize their cosine distance by using the CLIP model. It can be written as:

$$\mathcal{L}_t = 1 - cos\{E_I^{CLIP}(G(w')), e_t\}, \tag{10}$$

where $E_I^{CLIP}(\cdot)$ is the pretrained CLIP image encoder. Overall, the total loss is

$$\mathcal{L}_{\text{total}} = \lambda_{sp} \cdot L_{sp} + \lambda_t \cdot L_t, \tag{11}$$

where $\lambda_{sp}$ and $\lambda_t$ are set as 1 and 1.5 in our method, respectively.

## 4 Experiments

We first illustrate our implementation details. Then we compare FFCLIP with existing methods qualitatively and quantitatively. An ablation study validates the effectiveness of our modules. More results and a video demo are provided in the appendix and supplementary files, respectively. We will release our implementations to the public.

### 4.1 Implementation Details

We train and evaluate FFCLIP on the CelebA-HQ dataset [19], the LSUN cars dataset [55], and the LSUN Church dataset [55]. To invert images into StyleGAN2's latent codes, we use a pretrained e4e [46] as the image encoder. The dimensions of the latent code are $18 \times 512$, $16 \times 512$ and $14 \times 512$ for face, car and church images, respectively. When training our model, we leverage the text encoder from CLIP, and use the pre-trained StyleGAN2 to generate edited images. For each dataset, we use the corresponding pre-trained StyleGAN2 as the generator. We use 44 text prompts for face images containing emotion, hair color, hairstyle, age, gender, make-up, etc. Meanwhile, we follow the text prompts from StyleCLIP [33] when editing LSUN cars and church datasets. We randomly choose a text prompt for an input image for model training. In practice, we use a multi-step learning rate with an initial learning rate of 0.0005. The Adam [24] optimizer is utilized with $\beta1$ and $\beta2$ set to 0.9 and 0.999, respectively. For CelebA-HQ dataset, the total training iterations are 150,000 and the batch size is 8. For LSUN church dataset, the total training iterations are 200,000 and the batch size is 4. For LSUN cars dataset, the total training iterations are 100,000 and the batch size is 8. We train our model on a workstation with 8 Nvidia Telsa V100 GPUs.

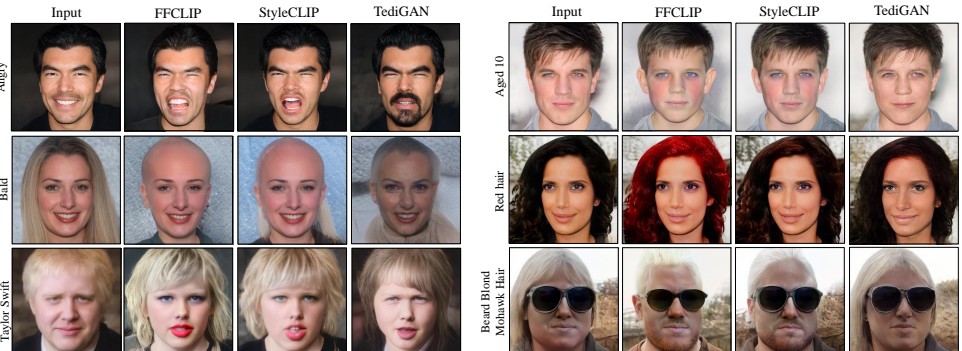

Figure 4: Visual comparison with TediGAN [53] and StyleCLIP [33] on the CelebA-HQ dataset. The text guidance is described on the left side of each row. FFCLIP is more effective to produce semantic relevant and visually realistic results.

## 4.2 Qualitative Evaluation

We compare our method to state-of-the-art text-guided image manipulation methods TediGAN [53] and StyleCLIP [33]. Then, we show our results with diversified text prompts. We also show our model is able to manipulate images with a single text prompt containing multiple semantic meanings.

**State-of-the-art Comparisons.** Fig. 4 shows the visual comparison results. We observe that the manipulated content does not exactly match the text prompt in the TediGAN's results. An example is shown on the first row with the 'Angry' prompt. The beard appears on the man in the TediGAN's result while it does not exists in the input image. While StyleCLIP improves the manipulation result of TediGAN, it still limits editing input images with high semantic accuracy. The hair still presents in the StyleCLIP result with the 'Bald' prompt. In comparison, our FFLIP is effective in editing images based on the text prompt semantics while maintaining visual realism. Compared to existing methods of training each model for one text prompt, our model processes multiple text prompts with only one model. Our superior performance results from the accurate semantic alignment across the text embedding and StyleGAN latent space.

**Free-form Image Manipulations.** Fig. 5 shows our image manipulation results where FFCLIP process text prompts with different semantic meaning. For human portrait images, FFCLIP preserves the face identity, produces high-quality edited images, and manipulates images with different semantics. Meanwhile, our results on cars and churches datasets are realistic with the accurate semantic transfer. The various manipulation results in multi datasets prove the robustness of our method. We pioneeringly develop a single model to process different text prompts for one type of images. More results are shown in Appendix.

**Text Prompts with Multiple Semantics.** We observe that although we train our model with single word text prompt, FFCLIP is able to edit images with text prompts containing multiple semantics. Fig. 6 shows the results. FFCLIP simultaneously transfers multiple text prompt semantics to the input image with realistic appearance. This success is because of our accurate semantic alignment for the text embedding from the CLIP model. More results are shown in Appendix.

**Interpolation Results.** FFCLIP can achieve fine-grained image manipulation by interpolating with two output latent codes. As shown in Fig. 7, we generate the intermediate latent code by linear weighting $w'_c = w'_a + \lambda(w'_b - w'_a)$, where $w'_c$ is the intermediate output, $w'_a$ and $w'_b$ are the outputs of FFCLIP which correspond to two different manipulation semantics. By gradually increasing the blending parameter $\lambda$ from 0 to 1, we can generate the results between two semantics (e.g., the visual results between 'Aged 10' and 'Aged 80'). More interpolation results are shown in Appendix.

## 4.3 Quantitative Evaluation

There is hardly a straight quantitative measurement to evaluate the image manipulation results. Nevertheless, in the text-driven image manipulation scenario, we believe the image results should

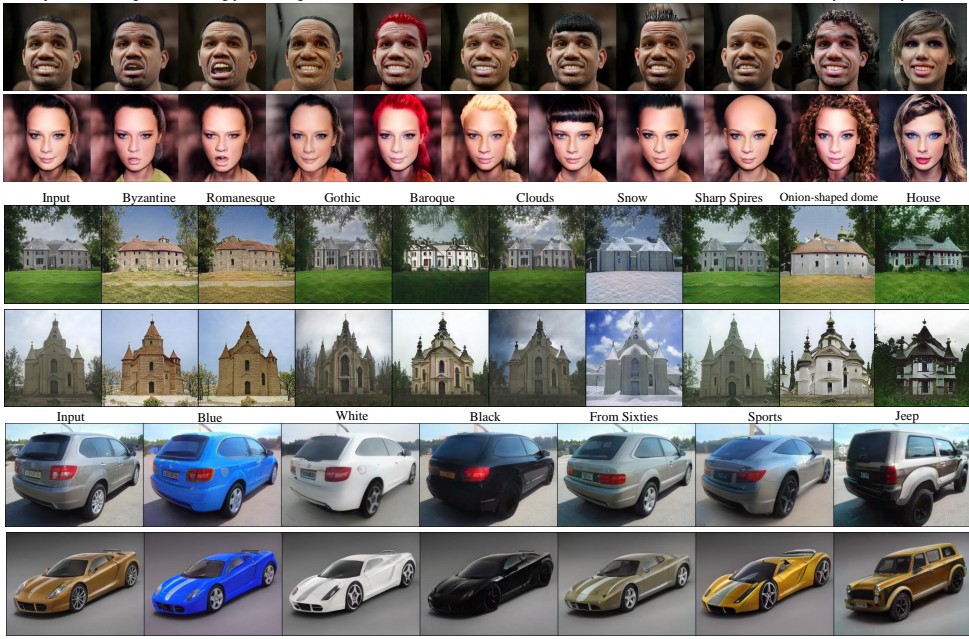

Figure 5: Our manipulation results on CelebA-HQ, LSUN cars and churches datasets with different text prompts. The input images are shown in the first column and our results are shown in the corresponding row. Our results are highly semantically relevant to the text prompts while maintaining visual realism.

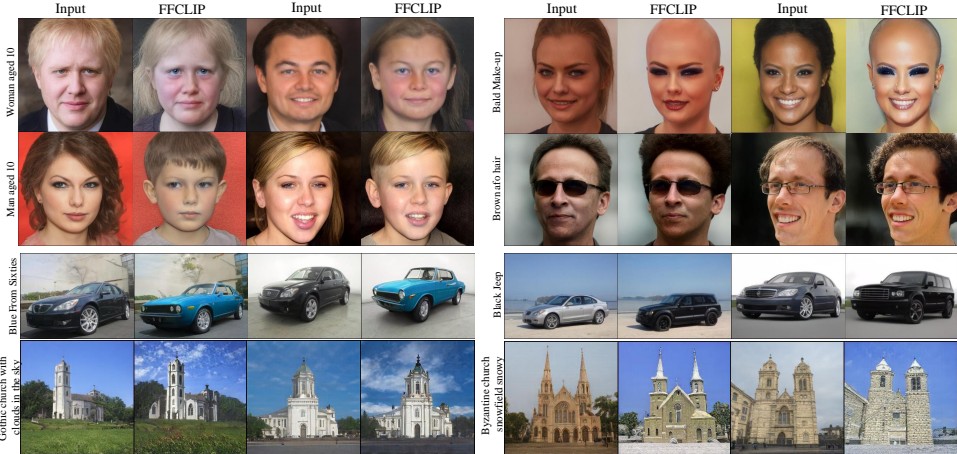

Figure 6: Our manipulation results with text prompts containing multiple semantic meanings. Our model, trained with text prompts consisting of a single semantic, is able to manipulate images based on multiple semantic meanings.

correspond to the text prompt semantics. We utilize several text prompts and randomly select 1000 testing images from the CelebA-HQ dataset. For each prompt, we produce the results from TediGAN, StyleCLIP and ours. Then, we follow [42] to use the multiple semantic classification models to measure the text-relevance of these results.

Table 1 shows these evaluation results under the editing performance column. We use different configurations to compare these results. When the text prompt is 'Bald', we use the PSPNet [59] to locate the hair region and count the number of pixels within this region. When the text prompt is 'Red hair', we compute the average color difference of the hair region of each result. When the text prompt is 'Angry', we use ESR [43] to determine whether the expression of the results belongs to the angry

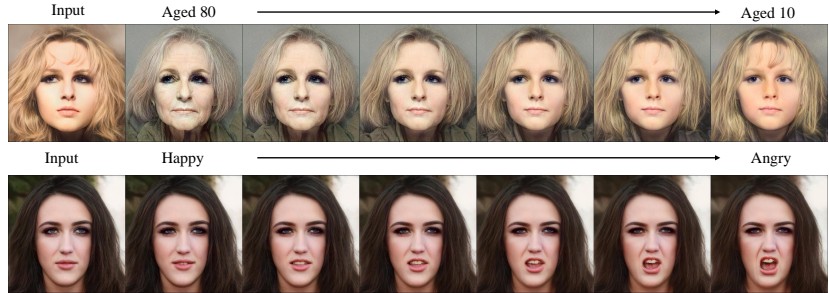

Figure 7: Interpolation results. We generate the intermediate results between 'Aged 80' and 'Aged 10' in the first row, and the results between 'Happy' and 'Angry' in the second row.

Table 1: Quantitative evaluations on the CelebA-HQ dataset. FFCLIP is more effective to produce semantically relevant results for both editing performance and human subject evaluation. '-' denotes that no quantitative comparison is given as no corresponding classifiers are available.

| Text Prompt | Editing Performance | | | Human Subject Evaluation | | |
|---|---|---|---|---|---|---|
| | TediGAN | StyleCLIP | Ours | TediGAN | StyleCLIP | Ours |
| Bald | 0.2507 | 0.1015 | **0.0279** | 5.7% | 8.6% | **85.7**% |
| Angry | 0.4520 | 0.4860 | **0.5810** | 0.0% | 17.1% | **82.9**% |
| Red hair | 1.0971 | 1.0416 | **0.7171** | 2.9% | 0.0% | **97.1**% |
| Aged 10 | 21.1448 | 17.0053 | **9.8874** | 2.9% | 0.0% | **97.1**% |
| Bowl cut hairstyle | - | - | - | 2.9% | 5.7% | **91.4**% |
| Beard blond mohawk hair | - | - | - | 8.6% | 0.0% | **91.4**% |

category. When the text prompt is 'Aged 10', we use an age classification method [39] to compute the distance between the estimated ages from the results and 10. For each prompt, we compute the quantitative numbers of all the results. As shown in this table, our method is more effective to transfer text prompt semantics to the images.

**Human Subjective Evaluation.** Besides designing different metrics to evaluate semantic transfer, we conduct human subjective evaluations on the manipulated results from compared methods. We randomly collected 54 images which were manipulated based on 6 text prompts. 35 participants with diverse backgrounds are asked to vote for the best results based on the three equally important principles. First, they should select the result where the semantic meaning corresponds to the text prompt most. Second, they should select the result where the human identity is best preserved. Third, they should select the most visually realistic image. We tally the votes as shown in Table 1 under the human subject evaluation column. It indicates that most participants favor our results compared to others, validating the effectiveness of our semantic modulation blocks. Moreover, we conduct other human subjective evaluations with more text prompts in Appendix.

## 4.4 Ablation Analysis

**Effect of Semantic Alignment Module.** The proposed semantic alignment module modifies Style-GAN's latent code $w$ from the position and channel dimensions by the scale and translation parameters. In order to verify the necessity of modulation with scale and translation, we trained three models: without scale parameter, without translation parameter, and without both parameters, respectively. It is worth mentioning that all three models contain a semantic injection module to inject text information into the latent space. As shown in Fig. 8, without scale or translation parameter, we cannot find the latent subspace for the text accurately (e.g., in row1 col4, 'bald' latent subspace is not found), and the latent subspace we find is not disentangled (e.g., in row1 col3, the face ID changes; in row1 col8, the hair color changes; and the result in row1 col9 wears glasses, etc.). Furthermore, only a complete model with both scale and translation parameters can be competent given a combination of texts unseen during training (e.g., beard blond mohawk hair in row2). The associated numerical results are shown in Table 2.

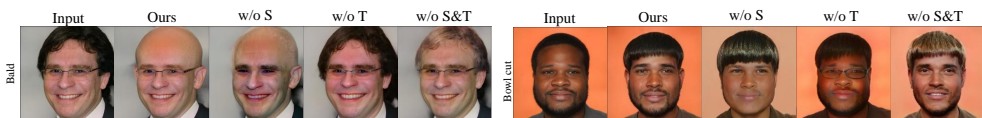

Figure 8: The effect of semantic alignment module. Input texts are given on the left, 'w/o S' means without scale parameter, 'w/o T' means without translation parameter, and 'w/o S&T' means without both. The results show that scale and translation modulation are essential for latent mapping.

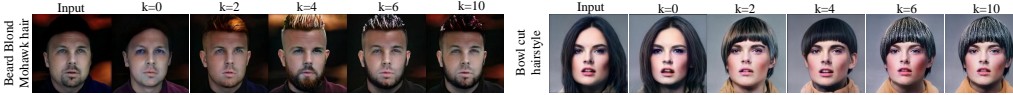

Figure 9: The effect of the number of semantic modulation blocks. We analyze 0, 2, 4, 6, and 10 semantic modulation blocks and choose to use 4 blocks.

Table 2: Quantitative ablation analysis results.

| Text Prompt | Editing Performance | | | | | | | | |
|---|---|---|---|---|---|---|---|---|---|
| | w/o S | w/o T | w/o S&T | Ours | $k=0$ | $k=2$ | $k=4$ | $k=6$ | $k=10$ |
| Bald | 0.0630 | 0.2391 | 0.2227 | **0.0279** | 0.2689 | 0.2371 | 0.0279 | **0.0236** | 0.1720 |
| Angry | 0.7510 | 0.7470 | 0.7640 | **0.5810** | 0.0669 | 0.5900 | 0.5810 | 0.7380 | 0.6206 |
| Red hair | 0.7082 | 0.7514 | 0.7769 | **0.7171** | 1.1566 | 0.7233 | 0.7171 | **0.6860** | 0.7515 |
| Aged 10 | 13.9412 | 20.0762 | 15.8426 | **9.8874** | 18.8625 | 18.5182 | 9.8874 | 13.4156 | 14.9789 |

**Effect of the Number of Semantic Modulation Block.** To ablate the number of modulation block ($k$ in Fig. 2), we train models with $k = 0, 2, 4, 6, 10$ respectively and evaluate their performance. The visual results are shown in Fig. 9. When we set the number of modulation blocks as 0, we use the mappers in StyleCLIP to train our model, and we find that the visual results cannot reflect the text semantics. A few modulation blocks will result in poor editing (see $k = 2$). Increasing the number of modulation blocks can improve the performance ($k = 6$), but larger $k$ will cause unstable training and lead to a higher computational cost (see $k = 10$). We finally select $k = 4$ in our method to balance the performance and efficacy. The numerical analysis results are shown in Table 2.

## 5 Concluding Remarks

We propose to manipulate image content according to text prompts with different semantic meanings. Our motivation is that latent mapping of existing approaches is empirically designed between text prompt embedding and the visual latent space, so one editing model tackles only one text prompt. We improve latent mapping by semantic modulations with alignment and injection. It benefits one editing model to tackle multiple text prompts. Experiments in multiple datasets show that our FFCLIP effectively produces semantically relevant and visually realistic results.

A limitation of our method is that we do not completely disentangle the StyleGAN latent code $\mathcal{W}+$, which still remains an open problem for all the existing methods. The CLIP is known to encode humanlike biases [52, 7]. As our text embedding is from CLIP, the manipulated results may also suffer from humanlike biases. Such biases may propagate and have negative effects on minority representations (i.e., skin color changes drastically after editing). To overcome this limitation, we can use the text-inversion [12] to find the proper text embedding without biases, we will try this in our future work. Moreover, FFCLIP generalizes when the training and testing text prompts share similar image attribute descriptions. It cannot change hair color if no hair-related prompts are utilized during training. The generalization of other encoders shown in Appendix remains for further investigations. The negative societal impact is that the edited human portraits could be used for malicious purposes.

**Acknowledgements and Disclosure of Funding.** We appreciate the insightful suggestions from the anonymous reviewers to further improve our paper. This work was supported by SZSTC Grant No.JCYJ20190809172201639 and WDZC20200820200655001, Shenzhen Key Laboratory ZDSYS20210623092001004 (Joint Research Center of Tencent and Tsinghua).

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
