# OpenReview forum: "One Model to Edit Them All: Free-Form Text-Driven Image Manipulation with Semantic Modulations"
_NeurIPS.cc/2022/Conference — NeurIPS 2022 Accept_

### Official Review · Reviewer_7Kr6 · 2022-06-16

**Rating:** 6
**Confidence:** 5
**Ethics Flag:** Yes
**Soundness:** 4 excellent
**Presentation:** 1 poor
**Contribution:** 3 good

**Summary:**

The paper proposes an attention-based mechanism for improving CLIP- based StyleGAN editing models.

Specifically, the paper takes the ‘latent mapper’ approach, where a neural network is trained to predict a latent-code change that would modify specific image attributes described through text. It argues that existing methods are trained to tackle specific texts, and therefore do not need to be aligned with CLIP’s space, limiting their ability to tackle novel texts.

The authors propose to better align the spaces through an attention mechanism, where the CLIP-text embedding is used to predict both layer and channel importance for latent code changes. This approach allows the authors to train their mapper on a larger collection of texts, creating better alignment between the CLIP and StyleGAN spaces, and enabling more disentangled modifications, with some generalization to unseen texts (e.g. combinations of texts seen during training).

**Questions:**

1) How well does the model generalize to unseen texts? Can a model not trained on hair-related prompts be used to change hair color? What if the only hair-related prompt was ‘blond hair’, would it still work to create red hair? What if the texts contained only ‘blond hair’ but also ‘red beard’, would compositionality in this manner work? If the model truly aligned the CLIP and StyleGAN spaces, one would expect this to be possible.

2) I’m a bit surprised that StyleCLIP fails so badly on hair color changes. Their official models have a purple hair mapper which works fine, and I just tried the global directions in their notebook and was able to generate ginger hair. Is there any chance that you trained their mapper with the fine-layers disabled?


**Ethics Review Area:**

["Discrimination / Bias / Fairness Concerns"]

**Limitations:**

The limitations and social impact section of the paper is extremely bare-bones, consisting of 2.5 lines.

A few likely limitations might be things discussed in my questions above: How well does the model generalize beyond the training text?
Another limitation might be the fact that you train specifically on e4e codes. What if better encoders come along and now, I want to use those? Can you generalize to latent codes derived through optimization? Using another encoder?

On the matter of social impact, this type of model can lead to harm in more than just the ability to create fake imagery. It can also propagate biases or have negative effects on minority representations. For example, in your Fig 6, the top-right individual’s skin color changes drastically after editing. It is fine that your model has these shortcomings, but the purpose of the limitations / impact paragraph is to be upfront about them.


**Strengths And Weaknesses:**

Strengths:
1) The paper introduces a conceptually simple yet well motivated approach for identifying the StyleGAN layers and channels most relevant to the change of specific attributes described through text – specifically, attention.
2) While there are prior works which deal with attention mechanisms in the context of StyleGAN inversion and editing [1, 2], they are, so far as I am aware, unpublished. However, I would urge the authors to cite these as related work (and consider comparisons to [2], which tackles the same task).
3) The results are generally of superior quality when compared to existing approaches. In particular, they are more disentangled, and allow for more complex manipulations than the baselines.

Weaknesses:

1) Clarity: The paper itself is very difficult to follow. I had a very hard time understanding the abstract and the introduction, and had to delve into the method and the experiments themselves to get any idea of what the authors are trying to do. The paper is not so complex that this should be required. I urge the authors to, at the very least, run their paper through Grammarly or similar tools.

2) Justification or experimental verification of claims: The core paper contains many claims which are unsubstantiated. In many cases, the supplementary **does** contain experiments which, at least in part, provide backings for these claims. However, they are never referenced from the core text, and if I did not read the appendix I would not be aware of them. These should absolutely be included in the main paper. A few examples:

    * The authors claim time and again that their model aligns the CLIP and StyleGAN spaces. The only experiment which possibly hints at this is the interpolation experiment in the appendix. Other than that, it appears that all experiments use the same, or compositions of the texts used during training.

    * It is unclear from the paper itself whether the benefit of their approach is a result of the attention mechanism (i.e. improved latent mapper architecture), or the fact that they train for multiple prompts. There is one experiment in the appendix that gives this some backing (Figure 7, the w/o S&T experiment). However, it would also be interesting to see that the original latent mapper can’t be trained conditionally on multiple prompts.


3) Some (minor) wrong claims:
   * In the paragraph of L107, the authors suggest that inversion is done into W+ because it is better for editing. This is not the case. Inversion is done into W+ because it allows for better identity preservation. W+ inversions typically behave worse under editing than their W counterparts. See for example e4e [3].
   * The authors claim that StyleCLIP needs to train a different model for each textual prompt. This is only true for their latent mapper approach, but is not actually specified anywhere in the current submission. The ‘Global Directions’ method, for example, does not suffer from this limit. Actually, I would clarify early in the paper that this method specifically aims to improve on the latent mapper approach of StyleCLIP (this also ties to clarity above).

[1] Transforming the Latent Space of StyleGAN for Real Face Editing, Li et al, 2021

[2] FEAT: Face Editing with Attention, Hou et al, 2022

[3] Designing an Encoder for StyleGAN Image Manipulation, Tov et al, SIGGRAPH 2021

---

> ### Author Response · Authors · 2022-08-02
> **Response to Reviewer 7Kr6**
>
> ***1. Related works.***
>
> Thanks for pointing this out. We have cited the mentioned two papers as related works. For the comparison to FEAT [15], we did not find available implementation. We will definitely compare FFCLIP to FEAT [15] when their implementation is accessible. On the other hand, we note that FEAT introduces an attention mask to edit the region of interests and follows the mapper of StyleCLIP. So FEAT still needs to train one model for one text prompt.
>
>
>
> ***2. Clarity.***
>
> We apologize for the confusion brought by our presentation. We have made significant changes in our abstract and introduction to present our observations and motivations, which give a clear picture of our objectives and intuitions. Meanwhile, we have thoroughly proofread our manuscript and fixed other unclear presentations and grammatical errors. Our modifications are marked in blue.
>
>
>
> ***3. Claim support.***
>
> We thank for your guidance on our paper organization, which makes our core text more supportive for our claims. We have revised accordingly and summarized our revision in the following:
>
> - In Sec. 4.2, we have shown the interpolation analysis in Ln 224-229 and Fig. 5. More results are provided in Fig. 14 and Fig. 15. To further support our latent mapping, we show our results produced by unseen text inputs in Fig. 19 and Fig. 20 in the appendix due to page limit. The results on unseen text inputs support our claim that our latent mapping is effective to align text embedding and latent visual space.
>
> - We have moved the ablation studies on semantic alignment module and number of semantic modulation block to the main text in Sec. 4.4. Moreover, we have tried training the original latent mapper from StyleCLIP by using multiple text prompts. We set k=0 in our experiment to present that we use the original latent mapper (see Ln. 270-273). The results are shown in Fig. 7 where the manipulated results do not faithfully present the semantic guidance from text prompts. The numerical analysis results are shown in Table. 2.
>
>
>
>
> ***4. Inaccurate claims.***
>
> - We agree that our illustration that inversion on W+ space benefits editing is not correct. We have replaced this claim by saying `W+ is more effective for identity preserving when we manipulate the image.`
>
> - We agree that StyleCLIP is a one for one scheme for their latent mapper approach while global directions do not suffer from this limit. We have added the illustration in Ln. 35-37 that the global direction method does not employ one for one training while its parameter adjustment and editing direction are manually predefined.
>
>
>
> ***5. Unseen texts.***
>
> Our FFCLIP is able to generalize to unseen text when the text prompts describe similar image attributes that have been utilized during training. If we use `blond hair` during training, we are able to produce manipulation results with `red hair` as input text prompt as shown in Fig. 20. On the other hand, we are able to produce results with `blond hair and red beard` as shown in Fig. 20. However, if FFCLIP is not trained on any hair-related prompts, it cannot change the hair color. More visual results with unseen text prompts are shown in Fig. 19 and Fig. 20 where FFCLIP is capable of generalizing if test text prompts and training text prompts describe similar image attributes (e.g., `blond hair` for training and `ginger hair` for test, `happy` for training and `rage` for test).
>
>
>
> ***6. StyleCLIP fails on hair color changes.***
>
> We follow the official setting of StyleCLIP and use the fine parts of latent code to train the mapper. We find that the StyleCLIP performs inferior under `Red hair`, while performing fine for other hair colors. As shown in Figure 25, we compare with StyleCLIP under `white hair` and `blond hair`. FFCLIP performs better than StyleCLIP in preserving unchanged facial attributes (e.g., for `white hair` on the first row, the facial skin color is changed in StyleCLIP. In the second row, the human mouth is closed in StyleCLIP. In the last row, the shape of eyeglass is changed in StyleCLIP). In contrast, our FFCLIP maintains such attributes while only changing the hair color. Also, under the `blond hair`, StyleCLIP produces hair color that resembles yellow rather than blond.
>
>
>
> ***7. Limitation discussions.***
>
> We thank you for the kind suggestions. Our FFCLIP generalizes when the training and test text prompts share similar image attribute descriptions. For different encoders, we have tried our model, which is learned with an e4e encoder to edit latent codes from High-fidelity [43] and Restyle [5] encoders, respectively, as shown in Fig. 21. Our FFCLIP performs well by using different inversion encoders. These results demonstrate that our FFCLIP can be potentially applied to different encoders.
>
>
> ***8. Social impact***
>
> We thank you for pointing out the minority representation concern and have added this aspect in Ln. 287-288.

---

> > ### Comment · Reviewer_7Kr6 · 2022-08-10
> > **Reviewer Response**
> >
> > I have read over the author rebuttal and the paper changes. These adequately address my main concerns, and so I am happy to raise my score accordingly.
> >
> > I believe clarity could still be improved, but the issue is not so severe that it should block possible publication.

---

### Official Review · Reviewer_rJnY · 2022-06-22

**Rating:** 8
**Confidence:** 5
**Soundness:** 4 excellent
**Presentation:** 3 good
**Contribution:** 4 excellent

**Summary:**

In this paper, an image manipulation method FFCLIP is proposed to edit image semantics based on the text prompt guidance. Following the StyleGAN latent space W for image generation, the alignment between semantics in the latent space W and text encoder is crucial for text-driven image editing. A semantic modulation method is developed to align the visual representations and text semantic embeddings via linear transformation where the parameters are computed based on cross attention computations. This effective alignment improves the previous empirical alignment design so that text prompts with different semantics can be taken for image editing via one model, while the empirical alignment confines each text prompt corresponds to one model. Experiments show that for each dataset (portraits, cars, or churches images), using one model is able to free-formly edit images based on different text prompts. The supplementary videos have also shown the effectiveness.

**Questions:**

See weakness. A clear discussion upon previous text-driven methods and more recent diffusion models would strengthen the proposed method.

**Limitations:**

As illustrated in Ln 271, the disentanglement of styleGAN latent space is an open problem. The negative society impact is for face malicious, which can be mitigated via deepfake detection methods.

**Strengths And Weaknesses:**

+ The alignment is effective to automatically correspond text prompt embeddings to StyleGAN latent Space. This enables different semantical embeddings to be reflected in the visual image representations for manipulations.
+ The semantic modulation first aligns text-image semantics via semantic alignment and then injects text prompts semantic. This design is reasonable to modify StyleGAN latent space since this space is not disentangled completely at present. Without latent space disentanglement, using the alignment via cross attention computation matches the text-image semantics.
+ Experiments on three benchmark datasets show that under each dataset, using one model is able to edit image semantics free-form based on different text prompts with single or multiple semantics. The visual results are impressive on both paper and the supplementary video. The numerical results also show the effectiveness of FFCLIP.

- The existing methods TediGAN, StyleCLIP, HairCLIP also edit images based on the text information. A more in-depth analysis shall be taken to enhance the proposed contributions. For instance, TediGAN seems to tackle image editing with different text attributes. StyleCLIP uses one model for a specific text-image editing. HairCLIP focus on hair and can handle different text semantics. These would strengthen why the proposed FFCLIP makes senses and contribute to the cross-modal image manipulation area.
- A discussion with diffusion models would be helpful to emphasize the importance of editing StyleGAN latent space to benefit future research.

---

> ### Author Response · Authors · 2022-08-02
> **Response to Reviewer rJnY**
>
> ***1. Discussions for StyleCLIP, HairCLIP, and TediGAN.***
>
> - The discussion between FFCLIP and StyleCLIP/HairCLIP has been presented in the first section of our response to `reviewer TtwX`. We suggest `rJnY` to check this section for reference.
>
> - In TediGAN, it makes the text feature close to the latent code and uses the style-mixing operation in StyleGAN to manipulate the image content. The style-mixing operation empirically replaces the specific layer in the latent code with text attribute embeddings. As these embeddings are not aligned well in the visual latent space, TediGAN is not effective to produce semantic abstraction results (e.g., see `Taylor Swift` of TediGAN in Fig. 4).
>
> - Moreover, TediGAN only manipulates the human portrait. This is because the empirical mapping between specific text attribute embeddings and specific layers in latent code is hard to reproduce in other datasets. The corresponding discussions have been added in Section A.5.
>
>
> ***2. Discussions with diffusion models.***
>
> Diffusion models have shown great potential in multimodal generation [29][30]. And the most relevant work to our FFCLIP is DiffusionCLIP [21]. DiffusionCLIP fine-tunes the reverse path with the CLIP loss and produces the noises to hijack the sampling process. In contrast to GAN, diffusion models have no semantic latent space (e.g., $W$, $W+$, or $S$ spaces), so it is not straightforward to generate the interpolation results. Meanwhile, the sampling process is time-consuming. Overall, diffusion models are a research direction, and we will take more investigations into them in the future.

---

> > ### Comment · Reviewer_rJnY · 2022-08-03
> > **Response to Author**
> >
> > I appreciate that the authors took the time to carry out the analysis I suggested. As I already indicated in my initial review, I still believe this is a very strong paper with many valuable contributions and insights.

---

> > > ### Author Response · Authors · 2022-08-03
> > > **Response to Reviewer rJnY**
> > >
> > > Thank you for your appreciation. I hope our work can help people better understand the semantic space of text and the latent space of GAN. Exploring the relationship between the two spaces can better promote the development of cross-modal generation.

---

> > > > ### Comment · Reviewer_rJnY · 2022-08-07
> > > > **Response to author**
> > > >
> > > > I think the author has solved all my doubts, and I think this paper deserves to be accepted by Nips, I am willing to accept this paper strongly.

---

### Official Review · Reviewer_TtwX · 2022-07-03

**Rating:** 5
**Confidence:** 4
**Soundness:** 2 fair
**Presentation:** 2 fair
**Contribution:** 2 fair

**Summary:**

This paper proposes a new architecture for using CLIP to modify images in coordination with StyleGAN. The system is known as FF-CLIP (Freeform CLIP) and incorporates semantic alignment and injection blocks to identify a semantic subspace in the GAN embedding space. Cross-attention is employed to achieve semantic alignment, while an injection technique previously used in HairCLIP is employed to inject the text embedding semantics into GAN space. The authors provide visual examples of image transformations and provide quantitative evaluations by human subjects indicating that humans prefer FF-CLIP modified images.

**Questions:**

Does model size of the CLIP used make any difference? The larger CLIPs do much better on evaluations like ImageNet, so maybe there would be some benefit to using them here?

**Limitations:**

CLIP is known to encode humanlike biases (see Wolfe et al. 2022, ACM FAccT; Birhane et al. 2021, arXiv). It would be good to mention these in a limitations section. How might they affect the system being developed?

**Strengths And Weaknesses:**

Originality
Strengths: The authors choose a relatively novel area of inquiry, that of providing text based guidance to a synthetic image generator, in this case a GAN. This is an area of much innovation at the moment.
Weaknesses: This seems very similar to other recent CLIP-based image modification systems, especially StyleCLIP. While the authors report outperformance of StyleCLIP in terms of human evaluation, it is difficult for this reader to tell that much of a difference between the StyleCLIP generated images and the FF-CLIP generated images based on the examples provided. The semantic injection technique has itself been used previously in HairCLIP.

Quality
Strengths: It is an interesting paper and to my knowledge the alignment/injection architecture is novel and might be useful.
Weaknesses: Evaluations are not robust enough to be convincing to this reader: the authors evaluate only six image modifications based on human preferences. It’s hard to know whether the results are significantly better than those achieved by, e.g., StyleCLIP when the evaluation is narrow.

Clarity
Strengths: Good visualizations, and the math is presented in a straightforward and comprehensible manner.
Weaknesses: The writing can be hard to follow, with grammatical issues throughout.

Significance
Strengths: Synthetic image generation is gaining ground rapidly and this system may help to advance the state of the art.
Weaknesses: The proposed system mainly consists of components previously used in very similar systems such as StyleCLIP and HairCLIP. If the evaluation of the system were more robust, this might not be an issue, but it is also hard to gauge how much of an advance this model is.

---

> ### Author Response · Authors · 2022-08-02
> **Response to Reviewer TtwX**
>
>
> ***1. FFCLIP is similar to CLIP-based image modification systems.***
>
> While we agree that both CLIP-based methods (i.e., StyleCLIP and HairCLIP) and our FFCLIP aim to solve the same task (i.e., text-driven image manipulation), we would like to show that the techniques between these methods are very different, which indicates the superiority of our FFCLIP.
>
> StyleCLIP performs an empirical mapping between each text prompt embedding and visual latent space of StyleGAN, so each StyleCLIP model is specifically trained to tackle one text prompt. As a result, the images produced in StyleCLIP based on different text prompt inputs are from different models.  In contrast, our FFCLIP establishes an automatic latent mapping (i.e., semantic alignment module) which enables the latent code modulation in visual latent space based on different text prompts. Thus, the results of our FFCLIP are from one single model for each type of image. The HairCLIP also performs an empirical mapping and only focuses on human hair regions, which is limited compared to our FFCLIP which generalizes well in multiple datasets, including human portraits, churches, and cars.
>
> We have emphasized these differences in Ln. 31-37, Section A.5, and Fig. 18 in the revised manuscript where existing CLIP-based systems are limited to establishing an automatic latent mapping between text embedding space and visual latent space. This limitation is addressed in our FFLIP model, tackling different text prompts for image manipulation.
>
>
>
> ***2. Visual difference between StyleCLIP and FFCLIP.***
>
> The accurate latent mapping of FFCLIP produces visual results highly related to text prompts while the empirical mapping of StyleCLIP does not. As shown in Fig. 4, given the `bald` input, the hair is completely removed in FFCLIP while it still exists in StyleCLIP. Given the `Red hair` input, the color of the hair in FFCLIP is obviously red while it is not the case for  StyleCLIP. More visual comparisons in Section A.8 show that FFCLIP produces images more related to the text prompt descriptions than StyleCLIP (e.g., given the `aged 10` as input, the result of FFCLIP is more like a child than that of StyleCLIP in Fig. 23).
>
>
>
> ***3. Six image modifications***
>
> As mentioned in Ln. 231-236, it is hardly to evaluate the performance of  image manipulation with  a straight quantitative measurement. We choose these six manipulation semantics in Table 1, since the text-relevance of these results can be measured by the classification model as mentioned in [36]. Meanwhile, we have provided thorough human subjective evaluations covering results produced based on four new text prompts('Blue eyes',  'Disgust',  'Dreadlocks hairstyle' and 'Jewfro hairstyle') in Table 4.
>
> Additionally, we have added more visual comparisons to HairCLIP in Fig. 16 and Fig. 17, as well as visual comparisons to StyleCLIP in Fig. 23, Fig. 24 and Fig. 25. These comparisons indicate FFCLIP is more effective to manipulate images given a wide range of text prompts.
>
>
>
> ***4. Writing***
>
> We apologize for the confusion brought by our presentation. We have thoroughly proofread our manuscript and fixed the unclear presentations and grammatical errors. Our modifications are marked in blue.
>
>
> ***5. Significance***
>
> As illustrated above, our automatic latent mapping between text prompt embedding and visual latent subspace empowers FFCLIP to tackle different text prompts while producing more semantically related results. Both visual and numerical comparisons shown above have indicated the effectiveness of FFCLIP.
>
>
>
> ***6. Model size of CLIP***
>
> In FFCLIP, we follow StyleCLIP to use the ViT-B/32 model to extract text prompt embeddings. We also try to use a larger model (ViT-L/14) and show the results in Fig. 22, which achieves similar performance. We attribute this to the reason that using a base model is sufficient to capture the text representations while using a model does not bring further advantages.
>
>
>
> ***7. Humanlike biases of CLIP.***
>
> We thank for this insightful suggestion and have added this analysis during in limitation discussion in Ln 285-288. As we use the text encoder from CLIP, the humanlike bias may appear in our visual results since our latent mapping connects text embeddings to visual latent space.

---

> > ### Comment · Reviewer_TtwX · 2022-08-09
> > **Response to Author Rebuttal**
> >
> > The authors have addressed many of my concerns, including adding more visual evaluations. I am increasing review score to 5.

---

> ### Author Response · Authors · 2022-08-07
> **Sincerely Look Forward to Your Feedback**
>
> Dear Reviewer TtwX:
>
> Thanks again for all of your constructive comments and suggestions, which have helped us improve the quality and clarity of this paper!
>
> We sincerely hope that our added experiments and analyses could address your concerns.
>
> Since the deadline for discussion is approaching, please feel free to let us know if there are any additional clarifications or experiments that we can offer, as we would love to convince you of the merits of our work. We appreciate your suggestions.
>
> Best wishes,
>
> Authors

---

### Review · Ethics_Reviewer_mGEh · 2022-07-31

**Recommendation:**

The application area, although also applied to cars, buildings, etc., is primarily focused on face images and will be difficult to think about if divorced from that application. The authors should most certainly, in detail, expound upon the ethical concerns in the document, but that is only a first step. Stronger guardrails should be considered, including bias mitigation as well as greater transparent documentation (cf. AI factsheets, datasheets, etc.).

**Ethical Issues:**

Yes

**Ethics Review:**

Image manipulation as a task has very foreseeable misuses in enabling fraud, surreptitious behavior, and other similar things. Moreover, the approach seems to be sometimes changing the ethnicity of the person in the input image, e.g. in Figure 1, Stephen Curry's ethnicity seems to have changed in the beard blond mohawk and aged 10 outputs. These sorts of effects are usually biased against minority groups. Remember the hoopla about the depixelator: https://www.businessinsider.com/depixelator-turned-obama-white-illustrates-racial-bias-in-ai-2020-6

---

### Review · Ethics_Reviewer_N3aQ · 2022-08-05

**Recommendation:**

The issue of not evaluating FFCLIP to measure its propensity to propagate harmful biases cannot be addressed in the existing paper.
It would require additional testing which would require substantial lead time and effort. A new version of the paper that included the test methods and results would be needed to address the issue.

**Ethical Issues:**

Yes

**Ethics Review:**

The solution presented leverages text embedding technology which is known to encode human biases.  The authors had not fully acknowledged the risks of propagating those risks and negatively impacting historically marginalized groups.  This risk was highlighted by 2  technical reviewers.

However, another issue not raised by the technical reviewers is that there was no attempt to adversarially test and evaluate the bias propagation properties of the solution.   The issues in this space are very well known and  acknowledging them is not enough. It should be regular practice  and mandatory to actually test and evaluate (there are a plethora of test sets and benchmarks that can be applied) solutions such as these for bias harms against historically marginalized groups and reporting them via something akin to a model card [Model Cards for Model Reporting, Mitchell M, et al 2019).

---

### Author Response · Authors · 2022-08-02
**General Reponse**

We sincerely appreciate all reviewers' efforts in reviewing our paper and giving insightful comments and valuable suggestions. We are glad to find that the reviewers generally acknowledge the following novelty and contributions of our work.

- Semantic Alignment: In contrast to the empirical text-visual semantic alignment, our semantic alignment module builds the relationship between text embedding in CLIP space and latent code in the latent space of StyleGAN, so that we can align the text space and latent space adaptively. This automatic alignment helps us to train one model to tackle different text prompts [TtwX, rJnY, 7Kr6].
- Experiments:  We evaluate our method in multiple datasets, including human portrait, car, and church images. The qualitative and quantitative analysis validated the effectiveness of our method [rJnY, 7Kr6].

As suggested by the reviewers, we have included the following contents in our revised manuscript to improve our paper. Our modification on the paper has been marked in blue. We summarize the major revision as follows. Our detailed responses can be found in the following response sections to the reviewers.

- More comparison results: We add more visual comparison results between FFCLIP and existing methods. Meanwhile, we add a new human subjective evaluation covering more text prompts to show the effectiveness of our method.[TtwX]
- Discussions on related works: We add the analysis of the difference between StyleCLIP, HairCLIP, TediGAN, and our method carefully. Meanwhile, we make the discussion for the diffusion model. [TtwX,rJnY]
- Unseen texts: To show the robustness of FFCLIP, we add visual results for the unseen texts. [7Kr6]
- Claim support: We adjusted our paper's structure, making the content better understood. [7Kr6]

---

### Meta-Review · Area_Chair_6UNu · 2022-08-28

**Recommendation:** Accept
**Confidence:** Certain

**Metareview:**

The paper develops an image manipulation method FF-CLIP (Freeform CLIP) to edit image semantics based on the text prompt guidance. A cross-attention module is developed to align the visual representations and text semantic embeddings. The results show the effectiveness of the approach. Reviewers had concerns on the novelty and comparison with previous similar image editing methods. Yet the reviewer-author discussion effectively addressed some of the concerns, and two reviewers raised their scores. The ethics reviewers expressed concerns on the potential ethics issues as one of the major applications of the work is human face editing, which is known to amplify biases. Though the revised version has added more discussion on the ethnics issue, more in-depth discussion/analysis on potential solutions would be desired.

**Award:**

No

---

### Decision · Program_Chairs · 2022-09-14

Accept